# Perturbative partial moment matching and gradient-flow adaptive importance sampling transformations for Bayesian leave one out cross-validation

## Abstract

Importance sampling (IS) allows one to approximate leave one out (LOO) cross-validation for a Bayesian model, without refitting, by inverting the Bayesian update equation to subtract a given data point from a model posterior. For each data point, one computes expectations under the corresponding LOO posterior by weighted averaging over the full data posterior. This task sometimes requires weight stabilization in the form of adapting the posterior distribution via transformation. So long as one is successful in finding a suitable transformation, one avoids refitting. To this end, we motivate the use of bijective perturbative transformations of the form $T(\boldsymbol{\theta}) = \boldsymbol{\theta} + hQ(\boldsymbol{\theta})$, for $0 < h \ll 1$, and introduce two classes of such transformations: 1) partial moment matching and 2) gradient flow evolution. The former extends prior literature on moment-matching under the recognition that adaptation for LOO is a small perturbation on the full data posterior. The latter class of methods define transformations based on relaxing various statistical objectives: in our case the variance of the IS estimator and the KL divergence between the transformed distribution and the statistics of the LOO fold. Being model-specific, the gradient flow transformations require evaluating Jacobian determinants. While these quantities are generally readily available through auto-differentiation, we derive closed-form expressions in the case of logistic regression and shallow ReLU activated neural networks. We tested the methodology on an $n \ll p$ dataset that is known to produce unstable LOO IS weights.

## 1 Introduction

In Bayesian workflows, multiple models are often fitted to a given dataset, and a selection procedure is applied to decide which model will be the most consistent with future observations. Prediction accuracy is most naturally estimated using cross-validation (CV) of which many variants exist. Commonly, a model trained using a given partition of the available data and evaluated using the remaining unused data. However, estimates of out-of-sample model metrics using train-test splitting are statistically noisy (Dietterich, 1998; Kohavi, 1995) unless computationally expensive k-fold cross-validation (i.e., fitting the model multiple times on cross-over the entire dataset) is employed (Rodriguez et al., 2010; Wong & Yeh, 2020).

Although $N$-fold – also known as leave one out (LOO) – CV is the most expensive of $k$-fold estimators, there exist computationally efficient LOO techniques that completely avoid refitting. For example, the Akaike Information criteria (AIC) and Bayesian variants (Stone, 1977; Watanabe, 2010; Gelman et al., 2014; Watanabe, 2013) are asymptotic approximations of LOO CV. For Bayesian models, a more precise way to compute LOO CV is to use importance sampling (Vehtari et al., 2017; Piironen & Vehtari, 2017a), which works by using the full data posterior measure as a proposal distribution for each data point's LOO posterior measure. However, in cases where the LOO measure and full measure are very different, importance sampling can fail (Piironen & Vehtari, 2017a). To ameliorate this possibility, we introduce an adaptive importance sampling methods for LOO CV based on using transformations that bring the proposal distribution closer to LOO posteriors under the principle that the transformation should be a small perturbation. We derive these transformations by defining gradient flows that minimize given statistical objective. While

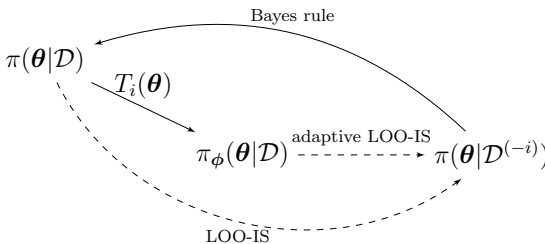

Figure 1: **Relationships between probability densities.** One wants to sample from $\pi(\boldsymbol{\theta}|\mathcal{D}^{(-i)})$, the LOO distribution for observation $i$, by sampling from the full-data posterior $\pi(\boldsymbol{\theta}|\mathcal{D})$. The transformation $T_i$ on the full-data posterior brings the sampling distribution closer to the target LOO distribution.

the transformations are model-dependent, the method is made completely general when using autograd for computing model gradients – for computational efficiency we derive the transformations exactly for a large class of classification models.

## 2 Preliminaries

### 2.1 Notation

We denote vectors (assumed to be column vectors unless otherwise stated) using bold-faced lowercase symbols, and matrices using bold-faced uppercase symbols. Given a matrix $\mathbf{W} = (w_{ij})$, the $i$-th row is denoted $\mathbf{w}_{i,}$ and $j$-th column is denoted $\mathbf{w}_{,j}$.

We refer to the entire set of observed training data as $\mathcal{D} = \{\mathbf{d}_i\}_{i=1}^n = \{(\mathbf{x}_i, y_i)\}_{i=1}^n$. As shorthand, we denote the set of training data where the i-th observation is left out as $\mathcal{D}^{(-i)} = \mathcal{D} \setminus \{\mathbf{d}_i\}$. Expectations with respect to the posterior distribution of $\boldsymbol{\theta}$ are denoted $\mathbb{E}_{\boldsymbol{\theta}|\mathcal{D}}$, and with respect to the posterior distribution of $\boldsymbol{\theta}$ if observation $i$ is left out are denoted $\mathbb{E}_{\boldsymbol{\theta}|\mathcal{D}^{(-i)}}$.

For a transformation $T : \Omega \to \Omega$, where $\Omega \subset \mathbb{R}^p$, we denote its Jacobian matrix $\mathbf{J}_T = \nabla T = \left(\partial_\alpha T^\beta\right)_{\alpha,\beta}$ and the determinant of the Jacobian matrix $\mathcal{J}_T = |\mathbf{J}_T|$. The gradient operator $\nabla$ operating on a function $\mu : \mathbb{R}^p \to \mathbb{R}$ is assumed to yield a column vector, the Hessian matrix for a function $\mu$ is denoted $\nabla\nabla\mu$, and the Laplacian of $\mu$ is denoted $\nabla^2\mu$

The operator $|\cdot|$ refers to determinants when the argument is a matrix, the 2-norm when the argument is a vector, and the absolute value when the argument is a scalar.

### 2.2 Importance sampling-based approximate leave one out cross validation (IS-LOO)

Suppose that one has pre-trained a Bayesian model such that one is able to sample its posterior parameters $\boldsymbol{\theta}_s \overset{\text{iid}}{\sim} \pi(\boldsymbol{\theta}|\mathcal{D})$. Our objective is to use knowledge of this full-data posterior distribution to estimate how the model would behave if any single point is left out at training. One can relate the full-data model to the model with observation $i$ left out using the Bayesian update equation

$$\pi(\boldsymbol{\theta}|\mathcal{D}) = \frac{\ell(\boldsymbol{\theta}|\mathbf{d}_i)\pi(\boldsymbol{\theta}|\mathcal{D}^{(-i)})}{\int \ell(\boldsymbol{\theta}|\mathbf{d}_i)\pi(\boldsymbol{\theta}|\mathcal{D}^{(-i)})\mathrm{d}\boldsymbol{\theta}}, \tag{1}$$

which is a Fredholm integral equation of the second-kind with respect to $\pi(\boldsymbol{\theta}|\mathcal{D}^{(-i)})$. This integral equation is in-practice difficult to solve due to the typically-high dimensionality of $\boldsymbol{\theta}$. Note that Eq. 1 is equivalent to Bayes' rule

$$\pi(\boldsymbol{\theta}|\mathcal{D}) \propto \pi(\boldsymbol{\theta}) \prod_{i=1}^n \ell(\boldsymbol{\theta}|\mathbf{d}_i). \tag{2}$$

Rather than directly inverting Eq. 1 to obtain $\pi(\boldsymbol{\theta}|\mathcal{D}^{(-i)})$, our starting point is the observation that Eq. 1 implies

$$\frac{\pi(\boldsymbol{\theta}|\mathcal{D}^{(-i)})}{\pi(\boldsymbol{\theta}|\mathcal{D})} = \frac{\mathbb{E}_{\boldsymbol{\theta}|\mathcal{D}^{(-i)}}[\ell(\boldsymbol{\theta}|\mathbf{d}_i)]}{\ell(\boldsymbol{\theta}|\mathbf{d}_i)} \equiv \nu_i(\boldsymbol{\theta}), \tag{3}$$

providing the ratio of densities functions between a distribution we know (the full-data posterior $\pi(\boldsymbol{\theta}|\mathcal{D})$) and a distribution whose statistics we would like to compute (the point-wise LOO posterior $\pi(\boldsymbol{\theta}|\mathcal{D}^{(-i)})$). To use the former to compute the latter we turn to Monte Carlo (Barbu & Zhu, 2020; Robert & Casella, 2013) – the use of statistical sampling to compute a desired quantity (typically an integral). Importance Sampling (IS) is a Monte Carlo method where one computes expectations with respect to a target distribution by taking a weighted average of samples with respect to a given proposal distribution. For an integrable function $f$,

$$\mathbb{E}_{\boldsymbol{\theta}|\mathcal{D}^{(-i)}}\left[f(\boldsymbol{\theta})\right] = \int f(\boldsymbol{\theta})\pi(\boldsymbol{\theta}|\mathcal{D}^{(-i)})\mathrm{d}\boldsymbol{\theta} = \int f(\boldsymbol{\theta})\frac{\pi(\boldsymbol{\theta}|\mathcal{D}^{(-i)})}{\pi(\boldsymbol{\theta}|\mathcal{D})}\pi(\boldsymbol{\theta}|\mathcal{D})\mathrm{d}\boldsymbol{\theta} = \mathbb{E}_{\boldsymbol{\theta}|\mathcal{D}}\left[f(\boldsymbol{\theta})\nu_i(\boldsymbol{\theta})\right]. \tag{4}$$

We approximate Eq. 4 by sampling over $\boldsymbol{\theta}_k \overset{\mathrm{iid}}{\sim} \pi(\boldsymbol{\theta}|\mathcal{D})$, and computing the Monte Carlo integral

$$\mathbb{E}_{\boldsymbol{\theta}|\mathcal{D}^{(-i)}}[f(\boldsymbol{\theta})] \approx \sum_{k=1}^{s} \nu_{ik} f(\boldsymbol{\theta}_k) \tag{5}$$

where the coefficients $\nu_{ik}$ are known as the self-normalized importance sampling weights

$$\nu_{ik} = \frac{\nu_i(\boldsymbol{\theta}_k)}{\sum_{j=1}^{s} \nu_i(\boldsymbol{\theta}_j)} = \frac{(\ell(\boldsymbol{\theta}_k|\mathbf{d}_i))^{-1}}{\sum_{k=1}^{s} (\ell(\boldsymbol{\theta}_k|\mathbf{d}_i))^{-1}}, \tag{6}$$

so that the undetermined constant $\mathbb{E}_{\boldsymbol{\theta}|\mathcal{D}^{(-i)}}\left[\ell(\boldsymbol{\theta}|\mathbf{d}_i)\right]$ cancels out. Eqs. 5, 6 define a well-known (Gelfand et al., 1992) Monte Carlo estimator for LOO.

## 2.3 LOO cross validation based metrics

The Bayesian LOO information criterion (LOO-IC), of which the Aikaike Information Criterion (AIC) is an asymptotic approximation, can be computed via:

$$\text{LOO-IC} = -2\sum_{i=1}^{n} \log \mathbb{E}_{\boldsymbol{\theta}|\mathcal{D}^{(-i)}}\left[\ell(\boldsymbol{\theta}|\mathbf{d}_i)\right] \approx -2\sum_{i=1}^{n} \log \sum_{k=1}^{s} \nu_{ik}\ell(\boldsymbol{\theta}_k|\mathbf{d}_i). \tag{7}$$

For classification problems, the out-of-sample area under the receiver operator curve or the precision-recall curve is often required. This can similarly be computed by propagating LOO estimates of the outcome probabilities

$$\widehat{p}_{\mathrm{loo},i} = \mathbb{E}_{\boldsymbol{\theta}|\mathcal{D}^{(-i)}}\left[p_i(\boldsymbol{\theta})\right] \approx \sum_{k=1}^{s} \nu_{ik} p(\boldsymbol{\theta}_k, \mathbf{x}_i). \tag{8}$$

## 2.4 Weight stabilization

Often it is the case that using the computed posterior $\pi(\boldsymbol{\theta}|\mathcal{D})$ as the proposal distribution for importance sampling has slow convergence properties – the $1/\ell$ importance weights, being fat tailed, are known to have large or unbounded variance (Peruggia, 1997), making the importance sampler estimate for LOO expectations (Eq. 6) noisy.

Two practical model agnostic methods for controlling the tail of importance weights are weight truncation (Ionides, 2008) and Pareto smoothing (Vehtari et al., 2024; 2017). Pareto smoothing replaces the largest $M$ weights with their corresponding rank-values from a fitted generalized Pareto-distribution (Zhang & Stephens, 2009). Pareto smoothed importance sample (PSIS)-based LOO implementations are widely available in software packages such as `Stan` and `ArviZ`. However, PSIS-LOO fails when the tail distribution of importance weights is not well-fit by the Pareto distribution; a general rule of thumb is that the parameter $\hat{k}$ exceeds 0.7. In these cases, performing an additional model-specific controlled transformation on the proposal distribution will induce more efficient computations. Later on, as in Paananen et al. (2021), we will use the estimated Pareto shape parameter $\hat{k}$ on post-transformation IS weights in order to evaluate the success of different transformations. Effective transformation should be able to reduce the Pareto shape parameter to below the given threshold.

## 2.5 Adaptive importance sampling

We apply the concept of adaptive importance sampling (Bugallo et al., 2017; Cornuet et al., 2011; Elvira & Martino, 2022) to transform the posterior distribution to be closer to the LOO distribution $\pi(\boldsymbol{\theta}|\mathcal{D}^{(-i)})$ (the relationships between the different distributions are depicted in Fig. 1).

Consider the bijection $T_i : \mathbb{R}^p \to \mathbb{R}^p$, defined for observation $i$, and let $\boldsymbol{\phi} \equiv T_i(\boldsymbol{\theta})$. By change of variables, $\pi_{\boldsymbol{\phi}}(\boldsymbol{\phi}|\dots) = \pi\left(T_i^{-1}(\boldsymbol{\phi})|\dots\right)\mathcal{J}_i^{-1}(\boldsymbol{\phi})$, where we denote $\mathbf{J}_T = \nabla T$, $\mathcal{J}_{T_i}^{-1}(\boldsymbol{\phi}) = \left|\mathbf{J}_{T_i}^{-1}(\boldsymbol{\phi})\right|$, and $\mathcal{J}_{T_i}(\boldsymbol{\theta}) = |\mathbf{J}_{T_i}(\boldsymbol{\theta})| = 1/\mathcal{J}_{T_i}^{-1}(\boldsymbol{\phi})$, The expectation in Eq. 4 in terms of an integral over $\pi_{\boldsymbol{\phi}}$ is

$$\mathbb{E}_{\boldsymbol{\theta}|\mathcal{D}^{(-i)}}\left[f(\boldsymbol{\theta})\right] = \int f(\boldsymbol{\theta})\nu_i(\boldsymbol{\theta})\pi(\boldsymbol{\theta}|\mathcal{D})\mathrm{d}\boldsymbol{\theta} = \int f(\boldsymbol{\theta})\nu_i(\boldsymbol{\theta})\frac{\pi(\boldsymbol{\theta}|\mathcal{D})}{\pi_{\boldsymbol{\phi}}(\boldsymbol{\theta}|\mathcal{D})}\pi_{\boldsymbol{\phi}}(\boldsymbol{\theta}|\mathcal{D})\mathrm{d}\boldsymbol{\theta}$$
$$= \int f(\boldsymbol{\theta})\nu_i(\boldsymbol{\theta})\frac{\pi(\boldsymbol{\theta}|\mathcal{D})\mathcal{J}_{T_i}(T_i^{-1}(\boldsymbol{\theta}))}{\pi(T_i^{-1}(\boldsymbol{\theta})|\mathcal{D})}\pi_{\boldsymbol{\phi}}(\boldsymbol{\theta}|\mathcal{D})\mathrm{d}\boldsymbol{\theta}. \tag{9}$$

Define a Monte Carlo approximation of Eq. 9 using importance sampling, by sampling $\boldsymbol{\theta}_k \overset{\mathrm{iid}}{\sim} \pi(\boldsymbol{\theta}|\mathcal{D})$ so that $\boldsymbol{\phi}_k = T_i(\boldsymbol{\theta}_k) \overset{\mathrm{iid}}{\sim} \pi_{\boldsymbol{\phi}}(\boldsymbol{\phi}|\mathcal{D})$ :

$$\mathbb{E}_{\boldsymbol{\theta}|\mathcal{D}^{(-i)}}\left[f(\boldsymbol{\theta})\right] \approx \sum_{k=1}^{s}\frac{\eta_{ik}}{\sum_{j=1}^{s}\eta_{ij}}f(\boldsymbol{\phi}_k) \qquad \eta_{ik} = \frac{\mathcal{J}_{T_i}(\boldsymbol{\theta}_k)}{\ell(\boldsymbol{\phi}_k|\mathbf{d}_i)}\frac{\pi(\boldsymbol{\phi}_k|\mathcal{D})}{\pi(\boldsymbol{\theta}_k|\mathcal{D})}. \tag{10}$$

By Bayes rule (Eq. 2), the posterior likelihood ratio in Eqs. 9–10 has the exact expression

$$\frac{\pi(\boldsymbol{\phi}|\mathcal{D})}{\pi(\boldsymbol{\theta}|\mathcal{D})} = \frac{\pi(\boldsymbol{\phi})}{\pi(\boldsymbol{\theta})}\prod_i\frac{\ell(\boldsymbol{\phi}|\mathbf{d}_i)}{\ell(\boldsymbol{\theta}|\mathbf{d}_i)}. \tag{11}$$

Computing this expression requires iterating over the entire dataset. There are various methods to avoid this expensive computation, for instance also using Monte Carlo approximation by sampling data points. For large datasets, one can turn to variational approximations.

## 2.6 Correcting variational posteriors

For computational expediency, variational methods are often used in place of MCMC for Bayesian inference, obtaining a variational approximation $\hat{\pi}(\boldsymbol{\theta}|\mathcal{D})$ to the true posterior, where $\hat{\pi}$ lies within a given family of probability distributions. In problems where one expects a substantial discrepancy between the true posterior and $\hat{\pi}$, one may correct for this discrepancy by noting that

$$\mathbb{E}_{\boldsymbol{\theta}|\mathcal{D}^{(-i)}}\left[f(\boldsymbol{\theta})\right] = \int f(\boldsymbol{\theta})\nu_i(\boldsymbol{\theta})\pi(\boldsymbol{\theta}|\mathcal{D})\mathrm{d}\boldsymbol{\theta} = \int f(\boldsymbol{\theta})\nu_i(\boldsymbol{\theta})\frac{\pi(\boldsymbol{\theta}|\mathcal{D})}{\hat{\pi}_{\boldsymbol{\phi}}(\boldsymbol{\theta}|\mathcal{D})}\hat{\pi}_{\boldsymbol{\phi}}(\boldsymbol{\theta}|\mathcal{D})\mathrm{d}\boldsymbol{\theta}$$
$$= \int f(\boldsymbol{\theta})\nu_i(\boldsymbol{\theta})\frac{\pi(\boldsymbol{\theta}|\mathcal{D})\mathcal{J}_{T_i}(T_i^{-1}(\boldsymbol{\theta}))}{\hat{\pi}(T_i^{-1}(\boldsymbol{\theta})|\mathcal{D})}\hat{\pi}_{\boldsymbol{\phi}}(\boldsymbol{\theta}|\mathcal{D})\mathrm{d}\boldsymbol{\theta} \tag{12}$$

and using the self-normalized importance sampler

$$\mathbb{E}_{\boldsymbol{\theta}|\mathcal{D}^{(-i)}}\left[f(\boldsymbol{\theta})\right] \approx \sum_{k=1}^{s}\frac{\chi_{ik}}{\sum_{j=1}^{s}\chi_{ij}}f(\boldsymbol{\phi}_k) \qquad \chi_{ik} = \frac{\mathcal{J}_i(\boldsymbol{\theta}_k)}{\hat{\pi}(\boldsymbol{\theta}_k|\mathcal{D})}\pi(\boldsymbol{\phi}_k)\prod_{j\neq i}\ell(\boldsymbol{\phi}_k|\mathbf{d}_j), \tag{13}$$

where $\pi(\boldsymbol{\phi}_k)$ is the prior density at $\boldsymbol{\phi}_k$, canceling out the two unknown constants corresponding to $\pi(\boldsymbol{\phi}_k|\mathcal{D})$ and $\nu_i$.

## 3 Methods

Eq. 10 is valid for an arbitrary bijection $T_i : \mathrm{supp}(\pi(\boldsymbol{\theta})) \to \mathrm{supp}(\pi(\boldsymbol{\theta}))$. The objective of using transformations is to shift the proposal distribution closer to the targeted LOO distribution for each observation – to invert

the update version of Bayes' rule. Returning to Eq. 1, in conjunction with Eq. 2, it is clear that the relative difference between the full posterior and the LOO posterior (for a given data point) is small – on the order of $1/N$. This fact motivates transformations of the form

$$T_i(\boldsymbol{\theta}) = \boldsymbol{\theta} + hQ_i(\boldsymbol{\theta}), \tag{14}$$

for a perturbation parameter $h > 0$ and function $Q_i$.

## 3.1 Partial moment-match stepping

The first two transformation methods presented in (Paananen et al., 2021) match the first moment and the first two moments respectively of the proposal distribution and the target distribution, independently for each scalar component of each model parameter. We generalize those transformations subject to a tunable scalar constant $\bar{h}$,

$$T_{\mathrm{PMM1}}(\boldsymbol{\theta}) = \boldsymbol{\theta} + \bar{h}(\bar{\boldsymbol{\theta}}_w - \bar{\boldsymbol{\theta}}) \qquad\qquad T_{\mathrm{PMM2}}(\boldsymbol{\theta}) = \boldsymbol{\theta} + \bar{h}\left(\mathbf{v}_w^{1/2} \circ \mathbf{v}^{-1/2} \circ (\boldsymbol{\theta} - \bar{\boldsymbol{\theta}}) + \bar{\boldsymbol{\theta}}_w - \boldsymbol{\theta}\right)$$

$$\bar{\boldsymbol{\theta}} = \frac{1}{s}\sum_{k=1}^{s} \boldsymbol{\theta}_k \qquad\qquad \mathbf{v} = \frac{1}{s}\sum_{k=1}^{s}(\boldsymbol{\theta}_k - \bar{\boldsymbol{\theta}}) \circ (\boldsymbol{\theta}_k - \bar{\boldsymbol{\theta}})$$

$$\bar{\boldsymbol{\theta}}_w = \frac{\sum_{k=1}^{s} \nu_k \boldsymbol{\theta}_k}{\sum_k \nu_k} \qquad\qquad \mathbf{v}_w = \frac{\sum_{k=1}^{s} \nu_k(\boldsymbol{\theta}_k - \bar{\boldsymbol{\theta}}) \circ (\boldsymbol{\theta}_k - \bar{\boldsymbol{\theta}})}{\sum_{k=1}^{s} \nu_k} \tag{15}$$

where setting $\bar{h} = 1$ recovers the original transformations MM1/MM2 respectively.

## 3.2 Gradient flow transformations

### 3.2.1 KL divergence descent

We consider choosing $T_i$ to minimize the KL divergence $\mathrm{D}_{\mathrm{KL}}\left(\pi(\boldsymbol{\theta}|\mathcal{D}^{(-i)})\|\pi_\phi(\boldsymbol{\theta}|\mathcal{D})\right)$, which is equivalent to minimizing the cross-entropy with respect to the mapping $T_i$,

$$H\left(\pi(\boldsymbol{\theta}|\mathcal{D}^{(-i)}), \pi_\phi(\boldsymbol{\theta}|\mathcal{D})\right) = -\int \nu_i(\phi)\pi(\phi|\mathcal{D})\log\frac{\pi(T_i^{-1}(\phi)|\mathcal{D})}{\mathcal{J}_{T_i}(T_i^{-1}(\phi))}\mathrm{d}\phi. \tag{16}$$

The Euler-Lagrange equation for minimizing Eq. 16 (derived in Supplemental Materials S.1.1), is implicit in $T_i$. While it admits no closed form solution, one may note that $T_i$ is a $t \to \infty$ stable fixed point of the KL-descending gradient flow

$$\frac{\partial T_i(\boldsymbol{\theta}, t)}{\partial t} = -\frac{\delta H\left(\pi(\boldsymbol{\theta}|\mathcal{D}^{(-i)}), \pi_\phi(\boldsymbol{\theta}|\mathcal{D})\right)}{\delta T_i} \tag{17}$$

where $\delta/\delta T_i$ denotes the functional derivative of $H$ with respect to the transformation $T_i$, and use this fact to refine, using the method of lines, an initial guess of $T_i(\boldsymbol{\theta}) = \boldsymbol{\theta}$ with forward Euler discretization of step size $h[\mathbb{E}_{\boldsymbol{\theta}|\mathcal{D}^{(-i)}}[\ell(\boldsymbol{\theta}, \mathbf{x}_i, y_i)]]^{-1}$, for $0 < h \ll 1$, to arrive at the transformation

$$T_i^{\mathrm{KL}}(\boldsymbol{\theta}) = \boldsymbol{\theta} - \frac{h}{\mathbb{E}_{\boldsymbol{\theta}|\mathcal{D}^{(-i)}}[\ell(\boldsymbol{\theta}|\mathbf{d}_i)]}\frac{\delta H\left(\pi(\boldsymbol{\theta}|\mathcal{D}^{(-i)}), \pi_\phi(\boldsymbol{\theta}|\mathcal{D})\right)}{\delta T_i}\Bigg|_{T(\boldsymbol{\theta})=\boldsymbol{\theta}} = \boldsymbol{\theta} + h\underbrace{\pi(\boldsymbol{\theta}|\mathcal{D})\nabla\left(\frac{1}{\ell(\boldsymbol{\theta}|\mathbf{d}_i)}\right)}_{Q_i^{\mathrm{KL}}} \tag{18}$$

### 3.2.2 Variance descent

In importance sampling, the variance of the estimator is conditional on the target function for expectation. Since we are interested in computing the LOO predictive probability for each observation $i$, it is natural to consider minimizing the variance of the transformed importance sampler for the function $p_i(\boldsymbol{\theta}) = p(\boldsymbol{\theta}, \mathbf{x}_i)$. However, this objective yields a transformation that is only useful for observations where $y_i = 0$ (see Supplemental Materials S.1.2). Instead, we seek to minimize the variance with respect to estimating the complement probability $p_i(\boldsymbol{\theta})^{1-y_i}(1 - p_i(\boldsymbol{\theta}))^{y_i}$.

Starting from the associated variational problem (Appendix S.1.2), and applying the same rationale that went into developing the KL-descending transformation, one arrives at the single-step variance-reducing transformation,

$$T_i^{\text{Var}}(\boldsymbol{\theta}) = \boldsymbol{\theta} + hQ_i^{\text{Var}}(\boldsymbol{\theta}) \qquad Q_i^{\text{Var}}(\boldsymbol{\theta}) = \pi(\boldsymbol{\theta}|\mathcal{D})\frac{f(\boldsymbol{\theta})}{\ell(\boldsymbol{\theta}|\mathbf{d}_i)}\nabla\left(\frac{f(\boldsymbol{\theta})}{\ell(\boldsymbol{\theta}|\mathbf{d}_i)}\right) \tag{19}$$

### 3.3 Resolving the posterior density

Both the KL (Eq. 18) and variance (Eq. 19) descent transformations take steps proportional to the posterior density $\pi(\boldsymbol{\theta}|\mathcal{D})$. If a variational approximation for $\pi(\boldsymbol{\theta}|\mathcal{D})$ is available, using it in Eqs. 18 and 19 as a stand-in for the posterior density helps simplify the computation of the transformations and their Jacobians, particularly when using mean-field or low-order Automatic Differentiation Variational Inference (ADVI) (Kucukelbir et al., 2017; Blei et al., 2017).

In the absence of variational approximation, one may evaluate the posterior densities exactly using the Bayes rule, absorbing the unknown normalization constant $\mathcal{Z}$ into the step size $h$. The obvious downside of using these exact transformations is the need to iterate over the entire dataset in order to evaluate the posterior density, which must be done for each parameter sample, for each data point.

For evaluating the Jacobian determinants, one appeals to Bayes rule to find that

$$\nabla\log[\mathcal{Z}\pi(\boldsymbol{\theta}|\mathcal{D})] = \nabla\log\pi(\boldsymbol{\theta}) + \sum_i\nabla\log\ell(\boldsymbol{\theta}|\mathbf{d}_i), \tag{20}$$

where $\mathcal{Z}$ is is absorbed into $h$.

### 3.4 Step size selection

The KL-divergence and variance descent transformations correspond to a forward Euler solver on the respective gradient flow equations. According to linear stability analysis, Euler's method has the conditional stability criteria $h < 2/\max_k|\text{Re}(\lambda_k)|$ where $\lambda_k$ are the eigenvalues of the Jacobian of the system (Jacobians of the functions $Q_i$). In each case the structure of the Jacobian admits inexpensive approximations of $\lambda_k$. However, for nonlinear systems, this criterion is not sufficient for achieving stability.

Instead, we use a modified rule to determine the step size. For all parameter samples at each individual observation $i$, we use

$$h_i = \bar{h}\min_{s,\alpha}\left\{\left|\frac{\sqrt{\Sigma_{\alpha,\alpha}}}{Q_i(\boldsymbol{\theta}_s)_\alpha}\right|\right\} \tag{21}$$

where $\bar{h} > 0$ and $\sqrt{\Sigma_{\alpha,\alpha}}$ is the marginal posterior standard deviation of the $\alpha$-th component of $\boldsymbol{\theta}$. This rule ensures that the transformation takes a step of at most $\bar{h}$ posterior standard deviations in any parameter component. The objective of adaptation is to find *any* transformation that results in importance weights where the Pareto tail shape is sub-threshold. To this end, one can compute the transformations for a range of $\bar{h}$ values in parallel using vectorized computations, saving computation at the cost of memory utilization.

### 3.5 Jacobian determinant approximation

For either single-step transformations, one may approximate $|\mathbf{J}_{T_i}|$ by noting that

$$\mathcal{J}_{T_i}(\boldsymbol{\theta}) = |1 + h\nabla\cdot Q_i(\boldsymbol{\theta})| + \mathcal{O}(h^2) \tag{22}$$

and truncating to $\mathcal{O}(h)$, sidestepping the computation of Hessian matrices and their spectra. Note that any higher order terms in this expansion require characterization of the spectra of $\nabla Q_i$, for each observation $i$, and for each sampled parameter $\boldsymbol{\theta}_k$. For large problems, computing the Jacobian matrix and its spectra many times can become computationally problematic.

### 3.6 Overview

We have presented four transformations, each aimed at stabilizing a LOO importance sampler by bringing the proposal distribution closer to the LOO target in a different sense. The PMM1/PMM2 transformations shift the moment of the posterior samples closer to that of the target distribution. The KL/Var descent transformations take one step along their corresponding gradient flow equations. While the latter two transformations use gradient information, their Jacobians are simple to approximate, requiring no computation of full Hessian matrices.

Generally, one will find that many observations are amenable to direct importance sampling with $1/\ell$ weights (Eq. 4) in combination with Pareto smoothing (tail weight distribution shape parameter $\hat{k} < 0.7$). One needs only transform the sampling distribution when the estimated shape parameter exceeds this threshold. For a given posterior sample of model parameters $\boldsymbol{\theta}_1, \ldots, \boldsymbol{\theta}_s \overset{\text{iid}}{\sim} \pi(\boldsymbol{\theta}|\mathcal{D})$, one undergoes for each observation $i$ the following algorithm:

> **procedure AdaptiveIS**(observation $i$)
>     Compute weights $\nu_{ik}$ (Eq. 6) and their tail shape $\hat{k}$
>     **if** $\hat{k} \leq 0.7$ **then Done**
>     **for** $T_i$ in transformations **do**
>         Apply $T_i$ to each $\boldsymbol{\theta}_k$
>         Compute weights $\eta_{ik}$ (Eq. 10)
>         Compute $\hat{k}$
>         **if** $\hat{k} \leq 0.7$ **then Done**

It is important to reiterate that if *any* transformation takes $\hat{k}$ for a given observation under the threshold then adaptation is successful – one avoids refitting the entire model.

## 4 Examples

Our focus is on classification models where a vector of covariates $\mathbf{x} \in \mathbb{R}^p$ is used to estimate the probability of an outcome labeled by $y \in \{0, 1\}$ with likelihood function $\ell$:

$$y_i|\boldsymbol{\theta}, \mathbf{x}_i \sim \text{Bernoulli}\left(p_i(\boldsymbol{\theta})\right) \qquad \ell(\boldsymbol{\theta}|y_i, \mathbf{x}_i) = p_i(\boldsymbol{\theta})^{y_i}(1 - p_i(\boldsymbol{\theta}))^{1-y_i}, \tag{23}$$

and where $p_i(\boldsymbol{\theta}) \equiv p(\boldsymbol{\theta}, \mathbf{x}_i)$ is the predicted outcome probability for observation $i$. In this manuscript, we pay special attention to the broad widely-used class of models that have a sigmoidal parameterization.

$$p_i(\boldsymbol{\theta}) = p(\boldsymbol{\theta}, \mathbf{x}_i) = \sigma(\mu_i(\boldsymbol{\theta})) \tag{24}$$

where $\sigma(\mu) = 1/(1 + e^{-\mu})$ is the sigmoid function and we denote $\mu_i(\boldsymbol{\theta}) \equiv \mu(\boldsymbol{\theta}, \mathbf{x}_i)$ for some mean function $\mu$.

For these models, the transformations take the form

$$Q_i^{\text{KL}}(\boldsymbol{\theta}) = (-1)^{y_i}\pi(\boldsymbol{\theta}|\mathcal{D})e^{\mu_i(\boldsymbol{\theta})(1-2y_i)}\nabla\mu_i \qquad Q_i^{\text{Var}}(\boldsymbol{\theta}) = (-1)^{y_i}\pi(\boldsymbol{\theta}|\mathcal{D})e^{2\mu_i(\boldsymbol{\theta})(1-2y_i)}\nabla\mu_i, \tag{25}$$

and their Jacobians take the form

$$\mathbf{J}_{T_i^{\text{KL/Var}}}(\boldsymbol{\theta}) = \mathbf{I} + \left\{ h(-1)^{y_i}\pi(\boldsymbol{\theta}|\mathcal{D})e^{(1+1_{\text{Var}})\mu_i(\boldsymbol{\theta})(1-2y_i)} \right.$$

$$\left. \times \left\{ \nabla\nabla\mu_i + [\nabla\log\pi(\boldsymbol{\theta}|\mathcal{D}) + (1 + 1_{\text{Var}})(1 - 2y_i)\nabla\mu_i](\nabla\mu_i)^{\intercal} \right\} \right\} \tag{26}$$

where

$$\nabla\log\pi(\boldsymbol{\theta}|\mathcal{D}) = \nabla\log\pi(\boldsymbol{\theta}) + \sum_j \left( y_j(1 - \sigma(\mu_j)) - (1 - y_j)\sigma(\mu_j) \right)\nabla\mu_j(\boldsymbol{\theta}), \tag{27}$$

and $\pi(\boldsymbol{\theta})$ is the prior. Here we will consider two popular subfamilies of sigmoidal models.

### 4.1 Logistic Regression (LR)

LR is a sigmoidal model where $\mu_i(\boldsymbol{\theta}) = \mathbf{x}_i^{\mathsf{T}}\boldsymbol{\beta}$, So, $\nabla_{\boldsymbol{\beta}}\mu_i = \mathbf{x}_i$, and $\nabla\nabla\mu = \mathbf{0}$. Because the Hessian of $\mu$ vanishes, the Jacobian of the function $Q_i$ for each of the functions is a rank-one matrix and has only a single non-zero eigenvalue. LR admits exact Jacobian determinants for each of the transformations:

$$\mathcal{J}_{T_i^{\mathrm{KL/Var}}}(\boldsymbol{\theta}) = \left| 1 + h(-1)^{y_i}\pi(\boldsymbol{\theta}|\mathcal{D})e^{(1+1_{\mathrm{Var}})\mu_i(\boldsymbol{\theta})(1-2y_i)}\mathbf{x}_i^{\mathsf{T}}\left[\nabla\log\pi(\boldsymbol{\theta}|\mathcal{D}) + (1+1_{\mathrm{Var}})(1-2y_i)\mathbf{x}_i\right]\right|. \tag{28}$$

### 4.2 Bayesian (ReLU) Neural Networks

Bayesian ReLU-nets (Lee, 2000; Ghosh & Doshi-Velez, 2017; Choi et al., 2018; Kristiadi et al., 2020; Bhadra et al., 2019) are piecewise linear (Sudjianto et al., 2021; Wang, 2022; Montúfar et al., 2014; Sudjianto et al., 2020) extensions to regression models. Being locally linear, these models have block-sparse Hessians and are also amenable to some limited degree of interpretability (Sudjianto et al., 2020; Chang et al., 2023). One may write an $L$-layer ReLU Bayesian neural network recursively

$$y_i|\mu_i \sim \mathrm{Bernoulli}(\sigma(\mu_i))$$
$$\mu_i|\mathbf{W}_L, b_L, \mathbf{z}_{L-1}^{(i)} = \mu(\mathbf{x}_i) = \mathbf{W}_L a(\mathbf{z}_{L-1}^{(i)}) + b_L$$
$$\mathbf{z}_k|\mathbf{z}_{k-1}^{(i)}, \mathbf{b}_k, \mathbf{W}_k = \mathbf{W}_k a(\mathbf{z}_{k-1}^{(i)}) + \mathbf{b}_k$$
$$\mathbf{z}_1^{(i)}|\mathbf{W}_1, \mathbf{x}_i = \mathbf{W}_1\mathbf{x}_i, \tag{29}$$

where $a$ is the ReLU activation function. The derivative of this function is the unit step function. We assume that the output function is sigmoid, noting that the softmax function also transforms into a sigmoid under a change of variables. Within the parameterization of Eq. 29 we absorbed the initial first-layer bias into the transformation $\mathbf{W}_1$, by assuming that $\mathbf{x}$ has a unit constant component, as is the convention in regression.

The Hessian matrix of $\mu$, while non-zero, is sparse because all of the following identities hold: $\nabla_{\mathbf{b}_k}\nabla_{\mathbf{b}_j}\mu = \mathbf{0}$ $\forall j, k$, $\nabla_{\mathbf{W}_k}\nabla_{\mathbf{W}_k}\mu = \mathbf{0}$ $\forall k$, $\nabla_{\mathbf{W}_k}\nabla_{\mathbf{b}_j}\mu = \mathbf{0}$ $\forall j \geq k$. For this reason, the Jacobian determinant approximation of Eq. 22 can ignore the model Hessian entirely. However, in the case of one hidden layer we exploit the Hessian's structure to provide explicit exact expressions for $\mathcal{J}_{(\cdot)}$.

**Example 4.1** (One hidden layer). These models are governed by the equations $\mu = \mathbf{W}_2 a(\mathbf{z}_1) + b_2$ and $\mathbf{z}_1 = \mathbf{W}_1\mathbf{x}$, where $\mathbf{W}_2 \in \mathbb{R}^{1\times d}$, $b_2 \in \mathbb{R}$, $\mathbf{W}_1 \in \mathbb{R}^{d\times p}$, $\mathbf{b}_1 \in \mathbb{R}^d$. This model has the first-order derivatives $\partial_{(W_2)_{1i}}\mu = a(z_1)_i$, $\partial_{(W_1)_{ij}}\mu = (W_2)_{1i}a'((z_1)_i)x_j$, $\partial_{b_2}\mu = 1$. The only non-zero components of the Hessian matrix for $\mu$ are the mixed partial derivatives

$$\frac{\partial^2\mu}{\partial(W_1)_{jk}\partial(W_2)_{1j}} = a'((z_1)_j)x_k. \tag{30}$$

The Hessian matrix of $\mu$ has a particular block structure that can be exploited (see Supplemental Materials S.2.1.1 for derivations) in order to find explicit expressions for its $2d$ non-zero eigenvalues, for $k \in \{1, 2\ldots, d\}$,

$$\lambda_k^{\pm} = \pm\left[\sum_j a'((z_1)_k)x_j^2\right]^{1/2}, \tag{31}$$

and associated eigenvectors

$$\mathbf{v}_k^{\pm} = \begin{pmatrix} \tilde{\mathbf{u}}_k/\sqrt{2|\mathbf{u}_k|^2} & \pm\mathbf{e}_k/\sqrt{2} & 0 \end{pmatrix}^{\mathsf{T}}, \quad \text{where} \quad \tilde{\mathbf{u}}_k = \begin{pmatrix} \overbrace{0 \quad \ldots \quad 0}^{(k-1)p \text{ zeros}} & \mathbf{u}_k^{\mathsf{T}} & \overbrace{0 \quad \ldots \quad 0}^{(d-k)p \text{ zeros}} \end{pmatrix}^{\mathsf{T}}, \tag{32}$$

and $\mathbf{u}_k = a'((z_1)_k)\mathbf{x}$. To compute the overall transformation Jacobians, one can then apply rank-one updates to $\nabla\nabla\mu$ – a process that is aided by projecting the model gradients into the eigenspace of the model Hessian (see S.2.1 for derivations).

## 5 Experiments

Jupyter notebooks for producing the results in the text are included in the Supplemental Results. As baselines for comparison, we evaluated the original MM1/MM2 affine transformation methods of Paananen et al. (2021), and the log-likelihood (LL) gradient descent method of Elvira et al. (2022) (derivations of this transformation for sigmoidal models are available in S.1). Note that in Paananen et al. (2021) they used a split-sampling scheme noting that all adaptations failed if it were omitted. In order to provide the most-direct comparison between the different transformations, we incorporated the MM1/MM2 transformations in the absence of split sampling.

### 5.1 Dataset and model

For demonstration, we used a public domain ovarian cancer micro-array dataset Hernández-Lobato et al. (2010); Schummer et al. (1999), consisting of $n = 54$ observations of $p = 1056 + 1$ predictors. As an example of a $p \gg n$ problem, model-agnostic $1/\ell$ importance sampling is insufficient for computing LOO expectations. Paananen et al. (2021) used this dataset to test their moment-matching adaptive importance sampler (in conjunction with split sampling) where they successfully decreased the number of observations where $\hat{k} > 0.7$ from approximately 35 to approximately 20 using $s = 1000$ posterior samples. We reproduced their logistic regression model, using the same regularized-horseshoe (Piironen & Vehtari, 2017b;c; Carvalho et al., 2009) prior, and the same statistical inference scheme within Stan, which we interfaced to Python using the package `cmdstanpy`. We ran four parallel Markov Chains, with twelve thousand burn-in iterations, retaining 2000 samples per chain (more details available in S.3.1). We then evaluated the transformation methods on resamplings of the retained MCMC samples.

### 5.2 Adaptation

| PMM1 | PMM2 | KL | Var | LL | MM1 | MM2 | LR ovarian | ReLUnet ovarian |
|------|------|------|------|------|------|------|------------|-----------------|
| - | - | - | - | - | - | - | $34.9 \pm 2.8$ | $16.8 \pm 3.7$ |
| ✓ | - | - | - | - | - | - | $5.3 \pm 1.7$ | $0.8 \pm 0.9$ |
| - | ✓ | - | - | - | - | - | $5.3 \pm 2.1$ | $0.1 \pm 0.3$ |
| - | - | ✓ | - | - | - | - | $17.9 \pm 2.8$ | $12.2 \pm 2.3$ |
| - | - | - | ✓ | - | - | - | $18.4 \pm 1.8$ | $16 \pm 3.4$ |
| - | - | - | - | ✓ | - | - | $22.2 \pm 3.2$ | $12.0 \pm 3.0$ |
| - | - | - | - | - | ✓ | - | $34.8 \pm 2.8$ | $9.8 \pm 2.9$ |
| - | - | - | - | - | - | ✓ | $34.9 \pm 2.8$ | $16.6 \pm 3.7$ |
| ✓ | ✓ | ✓ | ✓ | - | - | - | $0.4 \pm 0.5$ | $0.0 \pm 0.0$ |
| ✓ | ✓ | ✓ | ✓ | ✓ | ✓ | ✓ | $0.3 \pm 0.4$ | $0.0 \pm 0.0$ |
| | This work | | | | Comparisons | | | |

Table 1: **Counts of unsuccessful adaptations** (mean ± standard deviation) when using at least one of the given combination of transformations across the step sizes $\bar{h} \in \{4^{-r} : r \in \{0, 1, \dots, 10\}\}$, as seen in one hundred simulations of parameter sample size $s = 1000$. **Lower is better.**

We scanned different values of $\bar{h} = 4^{-r}$, for $r \in \{0, 1, 2, \dots, 10\}$, evaluating all transformations (KL/Var/PMM1/PMM2), and the comparison methods (LL/MM1/MM2) for a given value of $\bar{h}$. We performed this procedure 100 times, using samples of size $s = 1000$. Recall that adaptation is successful if *any* of the considered transformations can reduce $\hat{k}$ to below 0.7.

Table 1 presents statistics (mean ± standard deviation) for the number of observations where adaptation fails when using the given combination of methods. When using all methods in unison, one is generally able to successfully prevent the need to refit either the logistic regression or the neural network models for the task of obtaining LOO statistics. In particular the PMM1/PMM2 methods were highly effective for the RELUnet model.

For a representative instance of the simulation procedure in the context of logistic regression, Fig. 2 depicts the minimum value of $\hat{k}$ obtained for each transformation, organized by the index of each relevant observation

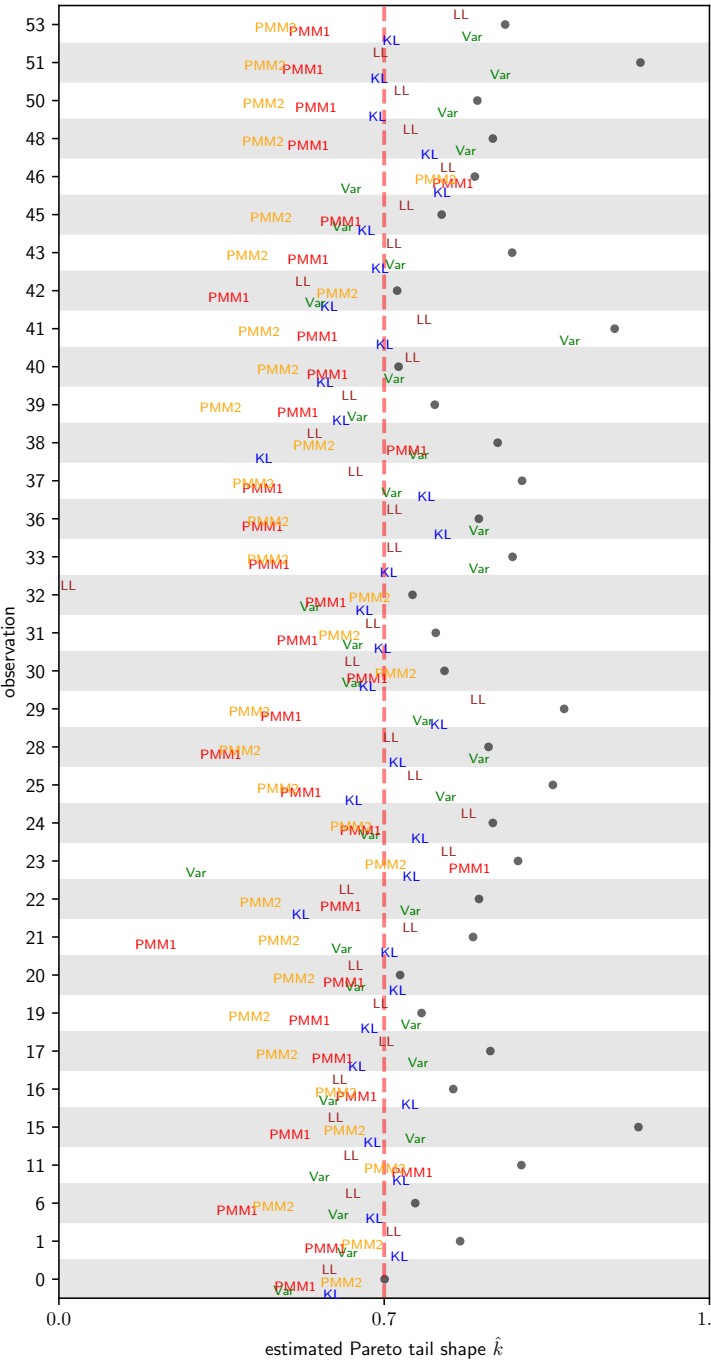

Figure 2: **Scatterplot of estimated Pareto tail shape diagnostic** $\hat{k}$ versus observation, for transformed ovarian cancer logistic regression model parameters, for observations where the untransformed samples have tail shape diagnostic $\hat{k} > 0.7$ (black dots •). Values of minimum $\hat{k}$ for each transformation plotted: green for KL, blue for Var, red for PMM1, orange for PMM2, purple for MM1, tan for MM2, and brown for LL – the minimum observed value for each transformation labeled. Adaptation for an observation is successful if $\hat{k} < 0.7$ for any transformation. If the minimum value for a given transformation and observation falls outside of this displayed range then the corresponding point is omitted from this plot.

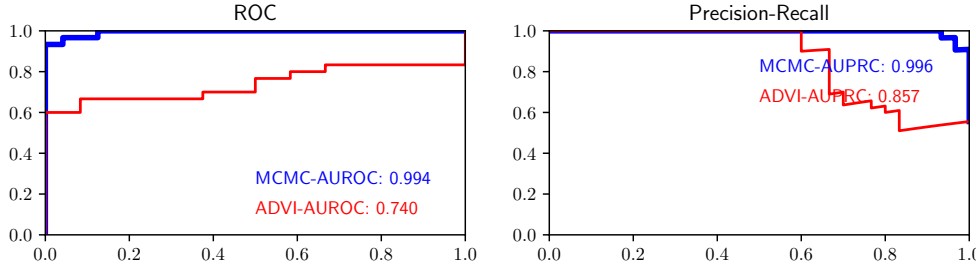

(a) **LOO ROC curves for logistic regression ovarian cancer classification models** contrasting the model fitted using MCMC and the model fitted using mean-field ADVI.

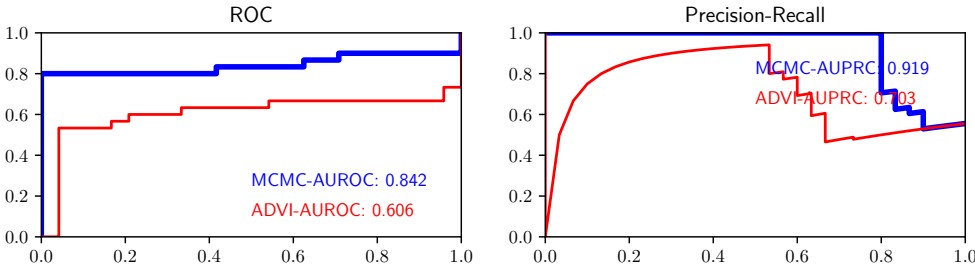

(b) **LOO ROC curves for neural network ovarian cancer classification models** contrasting the model fitted using MCMC and the model fitted using mean-field ADVI.

Figure 3: LOO ROC curves for ovarian cancer classification models fitted using MCMC and ADVI.

within the dataset. For each observation, the symbol • marks the pre-transformation value of $\hat{k}$. Generally, as seen in Fig. 2, PMM1/PMM2 had the most success. Hwoever, there are particular instances such as observations 23 and 46 where PMM1/PMM2 fail and the gradient flow transformations succeed in adaption. There are also many cases where multiple transformations each successfully adapt the posterior.

Fig. 3 shows the corresponding LOO- receiver operator characteristic (ROC) and precision recall curve (PRC) obtained by using the transformation that resulted in the best $\hat{k}$ value for each observation, predicting the LOO estimate of predictive probability, and feeding those probabilities into the relevant formulae for computing ROC and PRC. We contrast these curves for both MCMC and mean field ADVI -inferred variants of the model. The models inferred using MCMC have better generalization performance than their corresponding mean field approximations, which is not surprising due to the expected high degree of multicolinearity in this $p \gg n$ problem. This finding also held for the neural network model where it is notable that the MCMC-fitted neural network does not suffer as much from overfitting as does the mean field ADVI fitted neural network.

## 6    Discussion

In this manuscript, we introduced an adaptive importance sampler for using pre-trained full-data posteriors to approximate leave one out cross validation (LOO CV) in Bayesian classification models. The objective of importance-sampling LOO (IS LOO) is to compute LOO CV without incurring the computational cost of refitting a given model. The objective of adaptation to bring the sampling distribution (the full data posterior) closer to the target LOO posterior distributions for each data point so that IS LOO produces reliable estimates.

Our methodology is based on taking samples from the posterior and transforming them by adding a small step either towards the target expectation or according to the gradient flow corresponding to the minimization of a given objective. We introduce four such transformations: our PMM1/PMM2 generalizations of the MM1/MM2 transformations in Paananen et al. (2021) and the KL/Var gradient flow step transformations.

We presented explicit formulae for these transformations for logistic regression and ReLU-activated artificial neural networks. We described how one can easily approximate the Jacobian of the transformations for more-complicated models, including for ReLU neural networks of any size. The adaptive importance sampler is ultimately used to estimate the expected LOO prediction for each given datapoint – quantities that can be used to compute downstream model generalization metrics such as ROC/PRC curves and the area under these curves. Doing so without the need to refit a model saves considerable compute time and energy resources.

## 6.1 Contrasting and synergizing methods

Examining Table 1, taken individually, the KL and Var gradient flow-based transformations perform comparably to the original MM1/MM2 transformations (in unison with split sampling) evaluated in Paananen et al. (2021). Of note, as in Paananen et al. (2021), MM1/MM2 used in the absence of splitting was unable to successfully adapt any observation in our evaluations. However, the generalized PMM1/PMM2 transformations have by-far the best performance in shifting $\hat{k}$. Yet these two PMM methods, used either alone or together, usually do not completely get the job done. Each unsuccessful adaptation means that the model must be refit one additional time at high computational cost. Fortunately, using all the evaluated methods in-unison resulted in successful adaptation for all data points most of the time. The general strategy is then to loop through observations and try successive transformations for each observation until adaptation is successful.

## 6.2 Limitations

The main tradeoff of this method versus the model-agnostic PSIS-LOO method is that this method is model-dependent. In order to use this methodology for a given model, one needs to be able to evaluate gradients of the model with respect to parameters – and also the gradients of the corresponding prior distribution. Both the KL descent and variance descent transformations require computing the the posterior density – when a variational approximation of the posterior is not available or trustworthy this computation is costly for large datasets.

## 6.3 Extensions

In this manuscript, we focused on classification problems but the methodology for adapting the importance sampler is much broader. In the Supplemental Materials one may find more-general formulae for the KL and variance descending transformations. In medical and industrial contexts, one is often interested in whether an individual or unit will experience an outcome within a certain time interval. For instance, policymakers are interested in hospital readmission within 30 days post discharge (Xia et al., 2023; Chang et al., 2023) because these readmissions are possibly preventable. In these problems, one may apply survival modeling to characterize the lifetime distribution, and additionally evaluate a model according to its classification performance at a given cut-off time $T$. Our methodology can easily be used for assessing such models.

Another extension to this methodology is to take more steps along the gradient flow for a given objective. It may be feasible to learn such a transformation using neural or other expressive representations.

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
