# OpenReview forum: "Perturbative partial moment matching and gradient-flow adaptive importance sampling transformations for Bayesian leave one out cross-validation"
_TMLR — Rejected by TMLR_

### Review · Reviewer_Znq1 · 2025-07-15

**Summary Of Contributions:**

This paper presents novel adaptive importance sampling (AIS) techniques aimed at improving the stability and efficiency of Bayesian Leave-One-Out Cross Validation (LOO-CV). The authors introduce two new families of transformations to better approximate LOO posterior distributions without refitting the model: (1) Partial Moment Matching (PMM), a generalization of prior moment-matching schemes, and (2) Gradient Flow-based Transformations, derived from minimizing objectives such as KL divergence and variance of the IS estimator. Importantly, they provide closed-form gradient expressions for key models including logistic regression and shallow ReLU neural networks, and approximate Jacobian determinants for broader scalability.

Empirical results on a challenging high-dimensional dataset demonstrate that these transformations can often reduce the Pareto shape diagnostic parameter below the stability threshold (𝑘̂ < 0.7), thereby avoiding costly model refits. In particular, the PMM methods were especially effective on neural networks, and when combined with other methods, resulted in universal adaptation across observation

**Audience:**

Yes

**Claims And Evidence:**

Yes

**Requested Changes:**

* Author can discuss the potential implications on image datasets, particularly the pathology datasets where the number of observations is small compared to the gigapixel image predictors. This would greatly enhance the relevant claims on scalability.

**Strengths And Weaknesses:**

**Strengths**

* This paper is theoretically strong. Closed-form derivations of gradients and Jacobians for logistic regression and shallow ReLU networks are valuable for future implementations. The paper also effectively generalizes and extends existing MM1/MM2 transformations into more flexible PMM1/PMM2 schemes.

* Demonstrates notable performance on an unstable dataset (ovarian cancer microarray with n $<<$ p), successfully stabilizing importance weights where standard PSIS fails.

**Weaknesses**

* My major concern is the scalability. While the method is elegant for shallow models, the application to deeper or more complex neural networks (e.g., Bayesian transformer) is only partially addressed. Although Jacobian approximations are proposed (Eq. 22), deeper networks may involve high-dimensional parameter spaces where even first-order approximations become computationally intensive or unstable. The method's practicality for modern deep learning settings (e.g., foundation models) remains unclear.
*  Evaluation is limited to a single dataset and a narrow class of models (logistic regression and shallow neural networks). Demonstrating broader applicability (i.e., more datasets, such as pathology image datasets) across different model architectures (e.g., transformers, Bayesian CNNs) would strengthen the practical impact.

---

### Review · Reviewer_TLWU · 2025-07-19

**Summary Of Contributions:**

The paper considers Bayesian leave-one-out cross-validation, where one of the main challenges is the massive computational cost of refitting the model multiple times. Importance sampling can alleviate this problem; however, when the weights become large, the expected loss becomes noisy. To address this issue, the paper explores adaptive importance sampling, which designs transformations to move the posterior distribution closer to the target distribution. In this context, the paper contributes four new transformations. One is partial moment matching, which builds on existing methods to align the mean and variance of the target distribution. The other three are methods that transform the parameters by taking steps toward minimizing either the KL divergence or the variance of the estimator. On the presented dataset, the proposed methods are shown to successfully perform LOO, where other methods fail.

**Audience:**

Yes

**Broader Impact Concerns:**

I don't see any problem in this dimension.

**Claims And Evidence:**

Yes

**Requested Changes:**

Based on my overall positive assessment of the paper I don't feel like to request any change.

**Strengths And Weaknesses:**

Strengths:
- The paper introduces two new classes of adaptive importance sampling transformations.
- These two methods are practical, they don't have the high computational cost of refitting the model multiple times, and they are shown to perform well in practice.
- The exposition is clear.

Weaknesses:
- Based on my understanding, these methods require access to gradient with respect to model's parameters. Other "classical" methods are more general and model agnostic.

---

### Review · Reviewer_e8DF · 2025-08-09

**Summary Of Contributions:**

This paper proposes adaptive importance sampling strategies for LOO-CV. Building on existing work on adjusting the proposal distribution via invertible transformations of a specific form, the authors present several transformations that aim to adapt the full-data Bayesian posterior to approximate the LOO posterior. In particular, they introduce partial moment matching (PMM) and gradient-flow-based transformations, including KL divergence descent and variance descent. Closed-form expressions for the required quantities are derived for special cases such as logistic regression and ReLU networks.

**Audience:**

Yes

**Broader Impact Concerns:**

I dont fine any broader impact concerns for this submission.

**Claims And Evidence:**

No

**Requested Changes:**

- Please clearly state the technical novelty of the paper relative to prior work.

- Provide justification or theoretical support for the adaptive step-size rule.

- If possible, include empirical demonstrations on more complex models where only auto-differentiation (not closed forms) is available.

**Strengths And Weaknesses:**

Overall, the paper is well-written and easy to follow. However, I have several concerns that need to be addressed:

- Novelty: The general idea of employing adaptive IS for LOO-CV is well-established. The transformation form in Eq. (14) has been explored not only in the adaptive IS literature but also in other areas, such as particle optimization variational inference. The proposed PMM strategy appears to be a straightforward extension of existing moment matching methods, with the main change being the introduction of a step-size parameter—an extension that seems relatively natural and expected.

- Heuristic design choices: Several approximations and design decisions are presented without theoretical justification. For example, the adaptive step-size selection scheme in Sec. 3.4 is introduced as a computationally cheaper proxy for a theoretically grounded but more expensive method. However, no clear rationale is provided for its design or evidence for why it works reliably.

- Limited experimental scope: In the abstract, the authors note that “While these quantities are generally readily available through auto-differentiation, we derive closed-form expressions in the case of logistic regression and shallow ReLU activated neural networks.” However, all experiments are conducted on these models, for which closed-form simplifications exist. It remains unclear how the proposed approach performs for more complex, high-dimensional, and nonlinear models where closed forms are not available and the posterior may be multi-modal.

---

> ### Author Response · Authors · 2025-08-14
> **re: e8DF**
>
> # novelty
> The main novelty in our manuscript is that we unify two analytical ideas:
> - Transformations for LOO adaptation should be small in the sense of $|||T(\theta)-\theta||$ because LOO involves the removal of 1 out of the $N$ data points that went into training the posterior distribution for the model
> - Statistical costs (in our case the KL divergence and the variance) can guide the (direction of) the transformation
>
> Like most mathematical statements, they appear obvious in retrospect. However, these two ideas (to the best of our knowledge) have not been used in unison to motivate the development of AIS methods. For instance, the MM manuscript, in performing moment matching, is basing AIS on the latter of the two ideas. In our method we apply the former method in order to improve their method. Notably the MM manuscript tends to perform transformations that are too large. The modified PMM methods perform well but there are cases where they are not sufficient for adaptation.
>
> # step size
> Since we are performing a Bayesian procedure, the notion of a small transformation has a natural scale found in the variability of the posterior distribution. So the step size just needs to be tuned to yield transformations that are small relative to this scaling. Since the transformations themselves are cheap to compute, we believe that our scanning approach is sensible. We do not otherwise have any other apriori method to determine the step size, as we note in the limitations.
>
> # empirical evaluation
> We have since created a general port of the code using autograd so that one may use any likelihood function. It is possible for us to try an additional model but due to funding and employment issues we will likely not be able to perform these experiments for two or more months. We will note that it is not trivial to find a dataset that is good to demonstrate the different methods because most of the time the simplistic of the methods (PSIS smoothing for instance) are sufficient -- and often only very few data points need further methods for adaptation.

---

### Decision · Action_Editor_6Lsh · 2025-10-17

**Recommendation:** Reject

**Additional Comments:**

The authors are highly encouraged to revise and resubmit along the lines described above, as the reviewers were otherwise in favor of acceptance to TMLR.

**Audience:**

Yes

**Audience Explanation:**

All reviewers agreed that the findings would be of interest to the TMLR audience.

**Claims And Evidence:**

No

**Claims Explanation:**

(There were some delays in processing the decision for this submission due to the fact that the authors initially did not make their response viewable to 2 out of the 3 reviewers.) While some reviewers believe the claims in the submission are supported by accurate, convincing and clear evidence, one exception presents itself in the form of the review by reviewer e8DF. Specifically, this reviewer mentions that "the paper has limited experimental scope.... all experiments are conducted on these models, for which closed-form simplifications exist. It remains unclear how the proposed approach performs for more complex, high-dimensional, and nonlinear models where closed forms are not available and the posterior may be multi-modal." (At least one other reviewer mentions this limitation of the work as well.) The authors mention in their response to reviewer e8DF that they have made their code more easily implementable by other researchers on more complex models. However, the reviewer believes (and I agree) that "applying the proposed approach to a moderate-sized neural network is neither prohibitively time-consuming nor resource-demanding, and such an evaluation would considerably strengthen the work." Based on this, this criteria will only be met after the authors carry out this revision. It is fine if this takes a few months to resolve.

**Resubmission Of Major Revision:**

The authors may consider submitting a major revision at a later time.

---

> ### Author Response · Authors · 2025-11-05
> **Thanks**
>
> We will submit a revision in a couple months. Unfortunately we were affected by the US government shutdown.